# Effects of Anthropogenic Activities on *Sardinella maderensis* (Lowe, 1838) Fisheries in Coastal Communities of Ibeju-Lekki, Lagos, Nigeria

Temitope Adewale [1,*], Denis Aheto [1], Isaac Okyere [1], Olufemi Soyinka [2] and Samuel Dekolo [3]

[1] Africa Centre of Excellence in Coastal Resilience, Department of Fisheries and Aquatic Sciences, School of Biological Science, College of Agriculture and Natural Sciences, University of Cape Coast, Cape Coast P.O. Box 5007, Ghana; daheto@ucc.edu.gh (D.A.); iokyere@ucc.edu.gh (I.O.)

[2] Department of Marine Sciences, University of Lagos, Akoka 101017, Nigeria; osoyinka@unilag.edu.ng

[3] Department of Urban and Regional Planning, Lagos State University of Science and Technology, Ikorodu 101233, Nigeria; dekolo.s@lasustech.edu.ng

* Correspondence: adewale.temitope@stu.ucc.edu.gh

**Abstract:** Small-scale fisheries are significant sources of nutrition and livelihood globally. However, increasing anthropogenic activities in coastal areas of developing countries have threatened the sustainability of artisanal fisheries and species. Fisheries of *Sardinella maderensis*, towards the global stock of which Nigeria contributes 9% and which is a significant livelihood source in the coastal communities of Ibeju-Lekki, Lagos, faces sustainability threats. This research investigated the effects of anthropogenic activities on *S. maderensis* fisheries in the coastal areas of Ibeju-Lekki, Lagos, Nigeria. The study adopted a mixed-method approach involving qualitative and quantitative research methods. These included species identification, water quality analysis, land-use change analysis, field surveys, focus group discussions, and interviews. Genetic analysis of the fish samples from the study area revealed that the species had a mean of 98% similarity to *S. maderensis*. While major urban and industrial land use has increased by 175% in the last four decades, the catch per unit effort (CPUE) of *S. maderensis* declined monthly to 0.0072 kg/H between 2003 and 2019. Linear regression indicated that anthropogenic variables explained approximately 39.58% of the variation in the CPUE ($p < 0.001$, $R^2 = 0.40$). Water samples showed that heavy metal levels were above international limits, with high total petroleum hydrocarbon (TPH) pollution in all stations (27.56 mg/L–3985.40 mg/L). Physiochemical analysis of water samples indicated TDS levels higher than the acceptable limits (mean = 24,971.1 mg/L) and inadequate chlorophyll-a levels (mean = 0.01 µg/L). Hence, urgent strategies are required to mitigate anthropogenic threats through inclusive coastal management policies supporting resilient artisanal fisheries.

**Keywords:** small-scale fisheries; anthropogenic threats; *Sardinella maderensis*; land use change; remote sensing; pollution; Nigeria



## 1. Introduction

Fisheries support about 600 million people's livelihoods and supply 214 million tons of fish and 17% of animal protein consumption among the world's population [1,2]. Globally, fish, among other aquatic foods, are high-demand products, reaching a production value of USD 424 billion in 2020 and substantially contributing to many countries' gross domestic products (GDPs), alleviating poverty and fostering nutritional security [2]. In Nigeria, fish is a vital source of protein, and fisheries constitute a significant sector of the economy, contributing approximately 5.40% of the country's GDP [3]. A country report by the FAO [4] states that small-scale fisheries dominate fish production in Nigeria by contributing over 80% of Nigeria's total domestic fish production. Small-scale fisheries are prominent along the Nigerian coast, especially along the Ibeju-Lekki coastline, which extends for about

75 km of the total 180 km of the Lagos state coastline, contributing the highest percentage of fish caught with respect to other coastal sections [5]. The fisheries are multi-species fisheries with a dominance of *Sardinella* spp. and *Caranx* spp. [6–8]. Despite high catches of these species that have attracted local artisanal fishers and foreign nationals, there is limited knowledge of the effect of anthropogenic factors on the abundance indices of the *Sardinella* spp. on the Lagos coastline, which have some of the highest economic values to the fisherfolk [7].

*Sardinella maderensis* (Lowe, 1838) is commonly known as Madeiran sardinella or flat Sardinella and is locally called *Sawa*. It is a schooling pelagic fish from the *Clupeidae* family [9]. It has an elongated body with a variable depth, black or blue/green colouring, and silvery flanks. Its size is usually 20–25 cm, and it inhabits the near-surface of coastal waters, shoaling at the surface or the bottom, down to 50 m. It feeds on various small planktonic invertebrates, fish larvae, and phytoplankton. *S. maderensis* is presently found in 43 countries worldwide, with Africa dominating the global fish catch. Using a ten-year average (2008–2017), Nigeria is the third highest contributor of *S. maderensis*, being responsible for 9% of the species' global catch [10]. *S. maderensis* dominates small-scale marine fisheries and is the main species captured in Nigeria's coastal waters, providing livelihood sources, nutrition, and income for several poor coastal communities in Nigeria [4]. *S. maderensis* is one of Nigeria's most abundant and economically valuable coastal pelagic species [11,12]. It accounts for 69% of the fish caught by artisanal fishers in the Ibeju-Lekki locality [7].

In the last few decades, anthropogenic activities have increased along the Nigerian coastline due to rapid population growth linked to industrialisation [13,14]. This growth has resulted in significant human pressures on marine ecosystems and biological stocks [15]. Massive industrial activities and urban developments have threatened the pelagic fish populations due to the degradation of coastal habitats in which *S. maderensis* is endemic [12,16]. Moreover, industrial developments associated with the Lekki Free Trade Zone, including dredging and land reclamation for the construction of the seaport and the petrochemical refinery, have destroyed mangroves and coastal habitats crucial to *S. maderensis* [15]. Polluting effluents such as petroleum hydrocarbons and heavy metals from industries threaten water quality, causing a decline in ecosystem services and environmental sustainability [17–19]. In addition, inefficient fishing standards and illegal and unregulated fishing have negatively impacted the fisheries' sustainability, leading to significant changes in species composition and decreased catches [20,21]. Altogether, escalating anthropogenic pressures threaten the sustainability of small-scale fisheries and the livelihoods of Ibeju-Lekki communities.

A few studies have investigated the effects of anthropogenic activities on fisheries, emphasising the impacts on the broader coastal fisheries in Nigeria [22,23]. However, there is limited knowledge on how increasing anthropogenic activities affect *S. maderensis* fisheries and the livelihoods of fisherfolk in the coastal communities of Ibeju-Lekki. Filling the knowledge gap respecting the effects of anthropogenic activities on *S. maderensis* fisheries in Ibeju-lekki is imperative for building fishers' resilience and achieving the Sustainable Development Goals (SDGs) 1, 2, and 14. Moreover, findings from the study will provide empirical evidence on how anthropogenic activities affect *S. maderensis* fisheries and fishers' livelihoods in Nigeria. The outcome will inform policy interventions to mitigate pollution, habitat degradation, and over-exploitation, promoting resilience in small-scale fisheries that will sustain livelihoods and conserve biodiversity in Nigeria.

In its objectives, the study sought to confirm the identity of the *Sardinella* species exploited in Ibeju-Lekki fisheries using genetic and morphological techniques; analyse the land use and land cover changes over time using geospatial analysis; and assess water pollution levels and habitat degradation through water quality analysis. Anthropogenic factors were correlated with *S. maderensis* abundance to examine what relationships exist. The fisherfolks' perceptions of anthropogenic impact and vulnerability were elucidated, and strategies for mitigating anthropogenic threats and promoting resilient small-scale fisheries were recommended for adoption.

## 2. Literature Review

Globally, small-scale fisheries employ millions of fishers and are significant sources of nutrition, food security, and livelihood, catering for many people [24]. However, increasing anthropogenic pressures threaten the sustainability of many small-scale fisheries [25,26]. Major anthropogenic threats include pollution and habitat degradation from coastal developments and overfishing [27–29]. These human activities pose severe sustainability threats to small pelagic species, which support food security in West Africa [30]. These stresses damage breeding grounds, reduce productivity, and threaten important species like *S. maderensis* [31,32].

*S. maderensis* is a dominant species in Nigerian small-scale fisheries [4]; however, the species is rated by the International Union for Conservation of Nature (IUCN) as "vulnerable" [12]. According to Akintola and Fakoya [14], the availability of the Clupeids family, to which *Sardinella* spp. belong, has diminished over the last decade due to habitat destruction and overfishing. Furthermore, the existing literature on the spatiotemporal dynamics, demographic parameters, and abundance indices of *S. maderensis* have focused on other West African coasts, excluding Nigeria. These include areas of the Ivory Coast [33], Cameroon [34], Benin [35], Liberia [36], and Ghana [37]. Being a vulnerable species, understanding anthropogenic impacts on *S. maderensis* is crucial to its sustainable management and conservation efforts; hence, correct species identification and insight into its genetic diversity are essential for its adaptability and resilience to human-induced and environmental changes [38–40]. Moreover, measuring the anthropogenic effects on *S. maderensis* species has been constrained by limited stock assessments and species identification, which provide pivotal knowledge for fisheries management [41–43]. Despite calls by the Food and Agricultural Organization [44] for species-specific research on the *S. maderensis* species in Nigeria, this has remained elusive; therefore, this study undertakes species identification as a precursor to investigating anthropogenic threats to *S. maderensis* fisheries.

Lagos's extensive coastal area has undergone rapid transformation due to urbanisation, industrialisation, and population growth, creating complex environmental problems due to unpredictability and scale diversity [45]. The increasing demand for fish for human consumption has steadily increased the fishing efforts of small-scale fisheries in recent years [46]. While these escalating pressures threaten the sustainability of small-scale fisheries, only a few studies have elucidated localised risks and impacts, including the effects of anthropogenic pressures on the *S. maderensis* fisheries in Nigeria [8,14,47,48]. Hence, explicating localised threats will inform policies for the sustainability of the *S. maderensis* fisheries and fishers' livelihoods in Lagos's coastal waters.

## 3. Materials and Methods

### 3.1. Description of the Study Area

The study area is Ibeju-Lekki, a municipality (local government area) in Lagos State, Nigeria, as shown in Figure 1. Lagos State lies between latitude 6°20′ N to 6°40′ N and longitude 2°45′ E to 4°20′ E. The Lagos coastline stretches 180 km across the Atlantic Ocean, constituting 22.5% of Nigeria's 853 km coastline. Although Lagos is spatially the smallest state in Nigeria, it is the most densely populated, occupying a landmass of 3577 sq. km., of which about 786.94 sq. km. (22%) consists of lagoons and creeks [49,50]. Ibeju-Lekki, a leading fishing area in Lagos State, covers approximately 75 km of the Lagos coastline [6,7]. About 80 coastal and lagoon communities exist in Ibeju-Lekki, with small-scale fisheries being a significant source of livelihood [8,51].

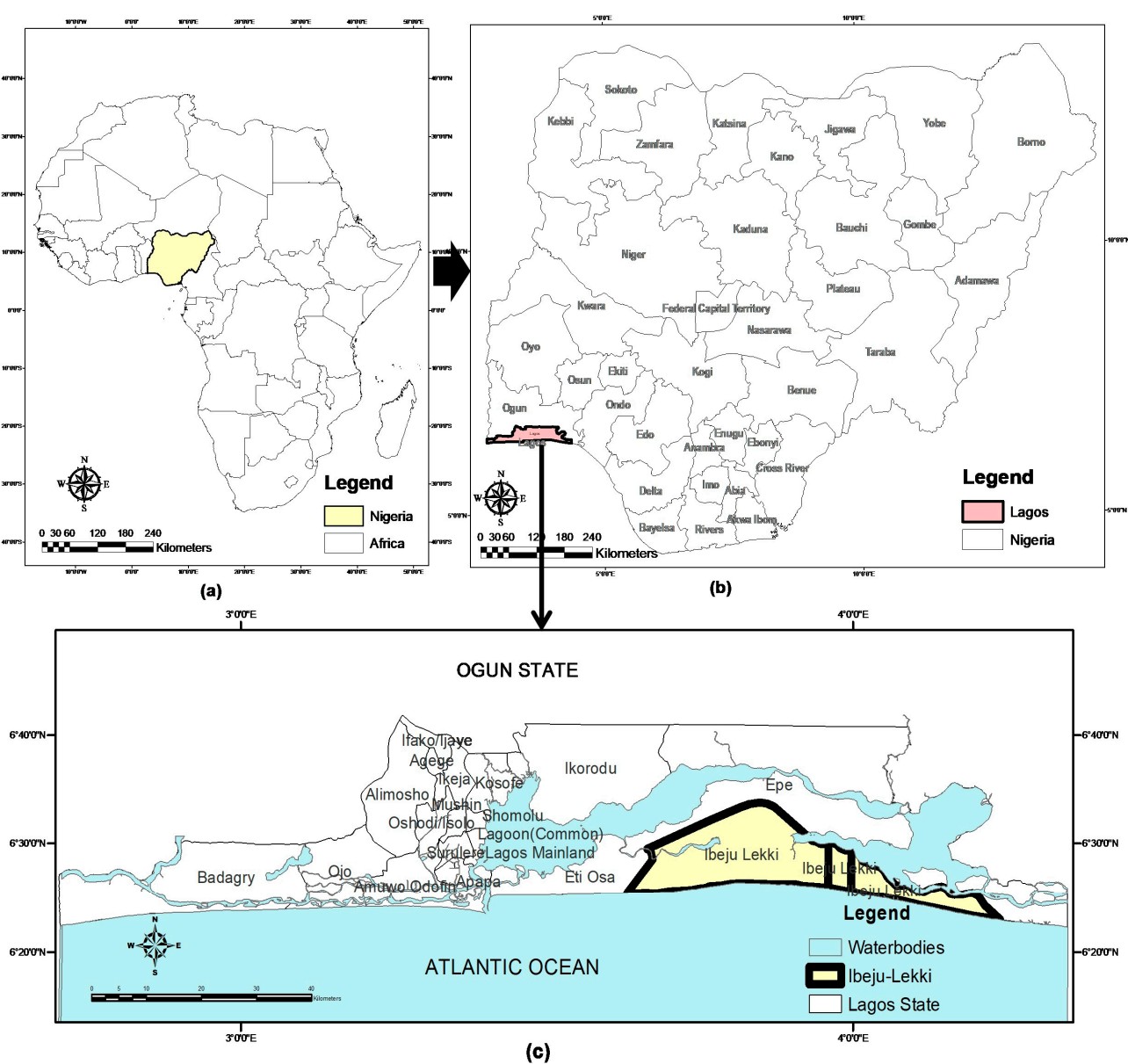

**Figure 1.** The geographical setting of the study area: (**a**) map of Africa showing Nigeria; (**b**) map of Nigeria showing Lagos State; (**c**) map of Lagos State showing Ibeju-Lekki.

### 3.2. Field Surveys

Field surveys were conducted to elicit responses from 30 coastal communities where *S. maderensis* fishing occurs in Ibeju-Lekki. Interviews, focus group discussions (FGDs), and observations were primary data collection methods used to derive perceptions of the targeted population of 1879 *S. maderensis* fishers in Ibeju-Lekki coastal communities (Table 1). Systematic random sampling was used to select 360 fishers proportionately across the 30 communities for interviews, while seven focus group discussions (FGDs) involving 6 to 10 fishers were conducted based on the principle of saturation [51]. Data collection instruments included a structured interview schedule and an FDG guide. Structured schedules contained variables on socio-demographics and fisherfolk's perceptions of vulnerabilities and effects of anthropogenic activities on their livelihoods. A test to validate the instrument returned a Cronbach's alpha value of 0.823, indicating the instrument's reliability. FGDs were used to obtain qualitative and more explicit responses to questions.

**Table 1.** List of fishing communities, *S. maderensis* fisher populations, and sample sizes.

| OID | Name of Community | X_Coord | Y_Coord | No. of Fishers | Respondents |
|---|---|---|---|---|---|
| 1 | Mopo Onibeju | 570649.904 | 710203.215 | 26 | 7 |
| 2 | Mosirikogo | 576039.373 | 710332.186 | 20 | 2 |
| 3 | Iwerekun | 580971.087 | 710695.66 | 22 | 5 |
| 4 | Igando Orudu | 588118.119 | 711854.276 | 20 | 4 |
| 5 | Debojo/Idado | 590680.321 | 712122.65 | 10 | 2 |
| 6 | Badore/Eleko | 592815.844 | 711999.131 | 6 | 1 |
| 7 | Magbon Alade | 598409.745 | 711667.942 | 290 | 52 |
| 8 | Orimedu | 600708.216 | 711570.328 | 315 | 57 |
| 9 | Orofun | 602640.31 | 711634.803 | 40 | 9 |
| 10 | Akodo | 603698.202 | 711494.404 | 185 | 34 |
| 11 | Tiye | 605610.807 | 711111.619 | 32 | 6 |
| 12 | Imobido | 606933.329 | 710870.107 | 12 | 1 |
| 13 | Idaso | 609293.271 | 710589.953 | 40 | 6 |
| 14 | Idotun/Magbon Segun | 613545.895 | 709805.322 | 40 | 8 |
| 15 | Okunraye | 616821.825 | 709262.975 | 20 | 5 |
| 16 | Olomowewe | 617879.415 | 709326.165 | 30 | 6 |
| 17 | Origanrigan | 618917.048 | 709206.33 | 15 | 3 |
| 18 | Oshoroko | 620422.918 | 708904.5 | 15 | 2 |
| 19 | Lekki | 621501.646 | 708601.792 | 60 | 7 |
| 20 | Apakin | 624818.66 | 707978.634 | 40 | 7 |
| 21 | Ita-Marun | 625958.063 | 707859.144 | 100 | 19 |
| 22 | Oriyanrin | 627240.777 | 707333.312 | 74 | 15 |
| 23 | Otolu | 628868.325 | 707255.64 | 80 | 15 |
| 24 | Okegelu | 630435.325 | 706974.531 | 40 | 14 |
| 25 | Lepia | 630924.571 | 706528.296 | 60 | 13 |
| 26 | Ikegun | 631900.825 | 706611.888 | 20 | 4 |
| 27 | Folu | 632633.442 | 706491.583 | 200 | 40 |
| 28 | OkunIse | 634138.993 | 706413.772 | 42 | 10 |
| 29 | AkodoIse | 634932.844 | 706212.301 | 15 | 6 |
| 30 | Imedu | 636683.217 | 705850.448 | 10 | 0 |
| | Total | | | 1879 | 360 |

### *3.3. Fish Species Identification*

Fish samples were collected from five out of the seven major landing sites shown in Figure 2, namely, Orimedu, Lekki, Magbon-Segun, Lepiya, and Folu, monthly between January 2021 and March 2022 and identified through morphological examinations conducted at the Marine Research Laboratory of the Department of Marine Sciences, University of Lagos, Nigeria. Biometric features of the fish species were examined, and identifications were completed using guidelines [52,53]. Key morphometric and meristic characteristics were examined to identify the *Sardinella* spp. species collected. Twelve (12) fish samples were used for the genetic analysis based on standards employed by Ward et al. [54], Ivanova et al. [55], and Kim et al. [56]. DNA was extracted from the fish fin tissues, and the cytochrome oxidase gene was sequenced using published primers [57]. The genetic analysis was performed by DNA extraction of the genomic DNA and PCR amplification using a synthesised primer for forward and backward reactions. The extracted fragments were sequenced using the Nimagen BrilliantDyeTM Terminator Cycle Sequencing Kit V3.1. An ABI350xl Genetic Analyzer was used to analyse the purified fragments. The ab1 files generated were edited using BioEditSequence Alignment Editor version 7.2.5, and a BLAST search in NCBI was conducted to obtain the results.

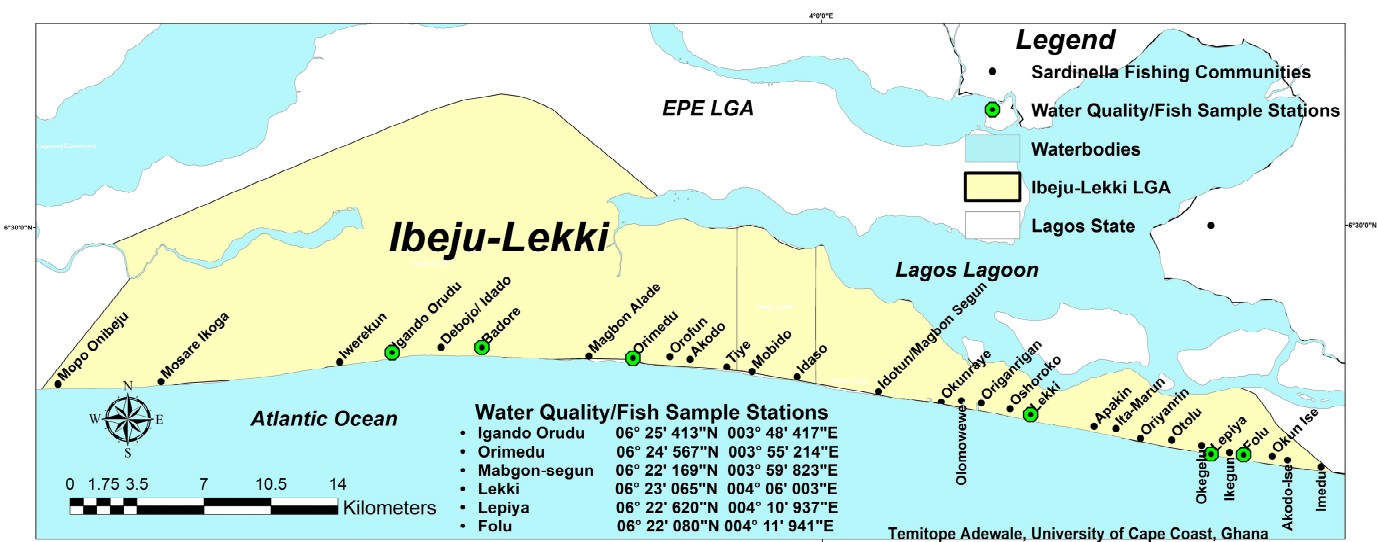

**Figure 2.** Map Ibeju-Lekki showing *S. maderensis* fishing communities and sampling stations.

*3.4. Land-Use and Land-Cover Change Analysis*

Landsat imageries were acquired from the United States Geological Surveys/Earth Resources and Observation Science (USGS/EROS) website and processed to determine the land use/land cover changes in Ibeju-Lekki coastal areas over about 36 years. As shown in Table 2, the satellite imageries used in this research include Landsat TM (Thematic Mapper) for 1984, which is the base year, Landsat ETM+ (Enhanced Thematic Mapper plus) for 2002, and Landsat OLI_TIRS (Operational Land Imager and Thermal Infrared Sensors) for 2020. The research adopted three (3) temporal periods based on Landsat imageries available to conduct a 36-year multi-temporal land-use change analysis from multi-spectral remote sense data for available periods (1984, 2002, and 2020). The spatiotemporal analysis provides concrete evidence to corroborate field data on environmental changes obtained from the field surveys in Ibeju-Lekki coastal areas.

**Table 2.** Spatial data and sources.

| Acquisition Date | Satellite Number | Sensor Type | WRS Path/Row | UTM Zone | Datum | Spatial Resolution (M) | Source and Year |
|---|---|---|---|---|---|---|---|
| 4 January 2020 | Landsat 8 | OLI_TIRS | 191/55 | 31 N | WGS84 | 30 | USGS, 2020 |
| 28 December 2002 | Landsat 7 | ETM+ | 191/55 | 31 N | WGS84 | 30 | USGS, 2006 |
| 18 December 1984 | Landsat 5 | TM | 191/55 | 31 N | WGS84 | 30 | USGS, 1984 |

False-colour RGB composite raster imageries (Bands 4,5,1 for Landsat 7 and Bands 5,6,1 for Landsat 8) were derived using ArcGIS Software version 10.3. A subset of the composite imageries limited to the AOI (area of interest—a 2 km buffer along the coastline) was extracted using a clip geoprocessing tool in ArcGIS (Figure 3). The RGB composite imageries were classified utilising the ISODATA unsupervised algorithm [58]. Imageries were synchronised with Google Earth, and ground truthing was performed to validate the classification. Change detection statistics were generated from land-use/land-cover change maps in TERRSET Software version 18.31.

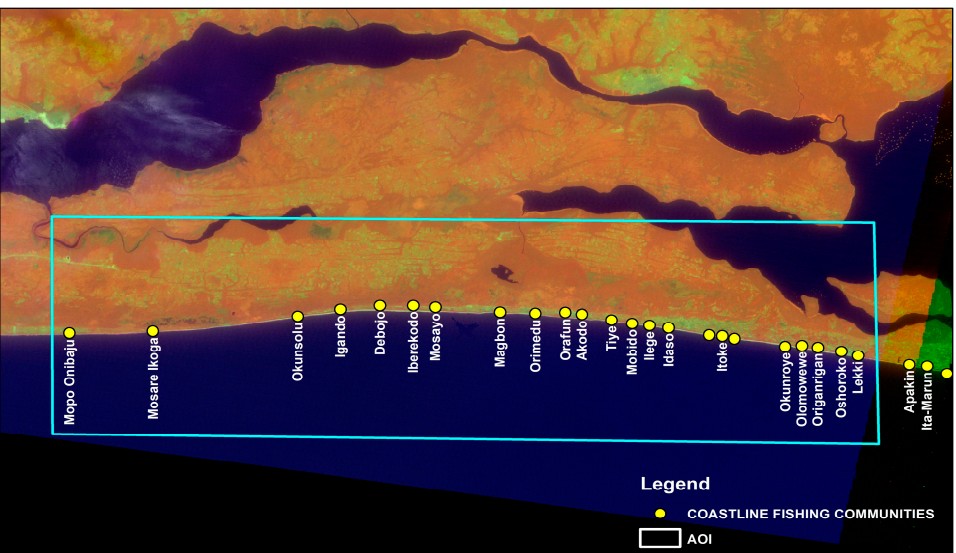

**Figure 3.** The 1984 false-colour RGB composites of Ibeju-Lekki (AOI in rectangle).

### 3.5. Water Quality Analyses

The water quality monitoring was performed bi-monthly at six major fish landing sites from February 2021 to January 2022 to allow for a yearly observational study of seasonal variations. Samples were collected using standard methods to analyse physical and chemical parameters like salinity, temperature, dissolved oxygen, nutrients, and heavy metals [59]. Total petroleum hydrocarbons (TPHs) were estimated through solid-phase extraction and gas chromatography [60]. The study used a solid-phase extraction (SPE) column separation method to remove organic and polar species from the water samples. The samples were eluted, evaporated, and reconstituted. The extracts were analysed for TPHs using a gas chromatograph fortified with a flame ionisation detector. Heavy metal analysis involved adding nitric acid to water samples, heating, cooling, filtering, and determining metal contents using the PG 990 Atomic Absorption Spectrophotometer.

### 3.6. Trends Analysis of S. maderensis Abundance

Monthly landing records for 2003 to 2019 were obtained for artisanal fisheries in Ibeju-Lekki from the Lagos State Agricultural Development Agency (LSADA) and the Federal Department of Fisheries (FDF) Nigerian repositories. Monthly landing records for *S. maderensis* fish catches in Ibeju-Lekki were obtained from LSADA and FDF records to calculate the monthly total catch, total efforts, and average catch per unit effort (CPUE). According to Arizi et al. [61] and Stobart et al. [62], catch per unit effort (CPUE) measures species abundance in fisheries. CPUE can be estimated by the quantity of the fish catch (weight) per unit of effort expended (time), which is proportional to the stock size [63]. Given an average of 7 h per trip, according to LSADA and the field survey, the CPUE was calculated using the formula:

$$\text{Average CPUE} = \frac{\text{Total Catch (in weight or numbers)}}{\text{Total Efforts (in hours)}} \tag{1}$$

Hence, having estimated the CPUE, monthly data compiled from 2003 to 2019 were used to generate a trend analysis of the CPUE of *S. maderensis* with a view to determining its implications for *S. maderensis* fisheries management and conservation efforts in Ibeju-Lekki. Trend analysis has been widely used to evaluate fish stock and fishing practices [64]. Also, to augment the observations on *S. maderensis* fisheries, fishers' local ecological knowledge (LEK) was used to understand anthropogenic factors responsible for the trend.

*3.7. Percieved Anthropogenic Effects on S. maderensis Abundance*

Collective perceptions held by fishers, also known as local ecological knowledge (LEK), offer crucial insights into trends in fish populations and anthropogenic impacts on small-scale fisheries, including fisheries management [65–67]. LEK, which adopts systematic and in-depth interviews of fishers, has been used to leverage the assessment of fish abundance in several studies [66,68,69], thereby giving room for inclusive fishery management [67,70,71]. Fishers' perceptions were obtained from interviews and analysed using SPSS version 25. Descriptive and inferential statistics were utilised to analyse fishers' perceptions of anthropogenic factors affecting *S. maderensis* fisheries and vulnerability. The dependent variable, CPUE, which represents *S. maderensis* abundance, and independent variables, such as fishing effort (V_FISH_EFFORT), land use effect (V_LANDUSE), amenity needs (V_AMENITY), and access to markets (V_MKT), were derived from observed variables, as shown in Table 3. Fish abundance (CPUE) was predicted by the perceived anthropogenic factors that were statistically significant using the linear regression model.

**Table 3.** Variable description.

| Variable | Description | Values | Measurement |
|---|---|---|---|
| V49 | Hours of fishing trip | {1–6 h, 7–12 h} . . . | Ordinal |
| V53a | Market/individual consumers | {No, Yes} | Nominal |
| V53b | Market/companies | {No, Yes} | Nominal |
| V53c | Market/middle women | {No, Yes} | Nominal |
| V54 | Average weight/quantity of each catch | {1–100 kg, 101–200 kg} . . . | Nominal |
| V61a | Land use effect/industrial dev. | {No, Yes} | Nominal |
| V61b | Land use effect/residential dev. | {No, Yes} | Nominal |
| V61c | Land use effect/recreational dev. | {No, Yes} | Nominal |
| V61d | Land use effect/commercial dev. | {No, Yes} | Nominal |
| V61e | Land use effect/transportation dev. | {No, Yes} | Nominal |
| V62a | Scale of effect/industrial dev. | {None, Least effect} . . . | Ordinal |
| V62b | Scale of effect/residential dev. | {None, Least effect} . . . | Ordinal |
| V62c | Scale of effect/recreational dev. | {None, Least effect} . . . | Ordinal |
| V62d | Scale of effect/commercial dev. | {None, Least effect} . . . | Ordinal |
| V62e | Scale of effect/transportation dev. | {None, Least effect} . . . | Ordinal |
| V67a | Needed amenities/good roads | {Not needed, Least needed} . . . | Ordinal |
| V67b | Needed amenities/hospitals | {Not needed, Least needed} . . . | Ordinal |
| V67c | Needed amenities/electricity | {Not needed, Least needed} . . . | Ordinal |
| V67d | Needed amenities/schools | {Not needed, Least needed} . . . | Ordinal |
| V67e | Needed amenities/telecom. facilities | {Not needed, Least needed} . . . | Ordinal |
| V67f | Needed amenities/pipe-borne water | {Not needed, Least needed} . . . | Ordinal |
| V67g | Needed amenities/financial incentives | {Not needed, Least needed} . . . | Ordinal |
| V67h | Needed amenities/extension services | {Not needed, Least needed} . . . | Ordinal |
| V61_LUE | Perceived land use anthrop. effect | {SUM: V61a, . . ., V61e} | Scale |
| V62_LUS | Scale of land-use anthrop. effect | {SUM: V62a, . . ., V62e} | Scale |
| V_LANDUSE | Perceived land use effects | {PRODUCT: V61_LUE, V62_LUS} | Scale |
| V_CATCHTRIP | Catch per trip (kg) | {V54} | Scale |
| V_FISH_EFFORT | Fishing effort (hours) | {V49} | Scale |
| CPUE | Catch per unit effort (kg/h) | {V54/V49} | Scale |
| V53_MKT | Access to market | {SUM: V53a, V53b, V53c} | Scale |
| V_AMENITY | Amenities (needed) | {SUM: V67a, . . ., V67h} | Scale |

## 4. Results

*4.1. Fish Species Identification*

The morphometrics-based identification system involving external or phenotypic features, including body shape, fin rays, and meristic counts, to identify the fish species revealed that the *Sardinella* spp. species exploited in Ibeju-Lekki are mainly *S. maderensis* (Figure 4). The fish showed features described by Whitehead [9] and Gourene and Teguels [72]. They had elongated bodies with a belly fairly sharply keeled, 18–23 dorsal

soft rays and 17–23 anal soft rays, 70–166 lower gill rakers, and upper pectoral fin rays that were white on the outer side, and the membrane was black [9,52,72,73].

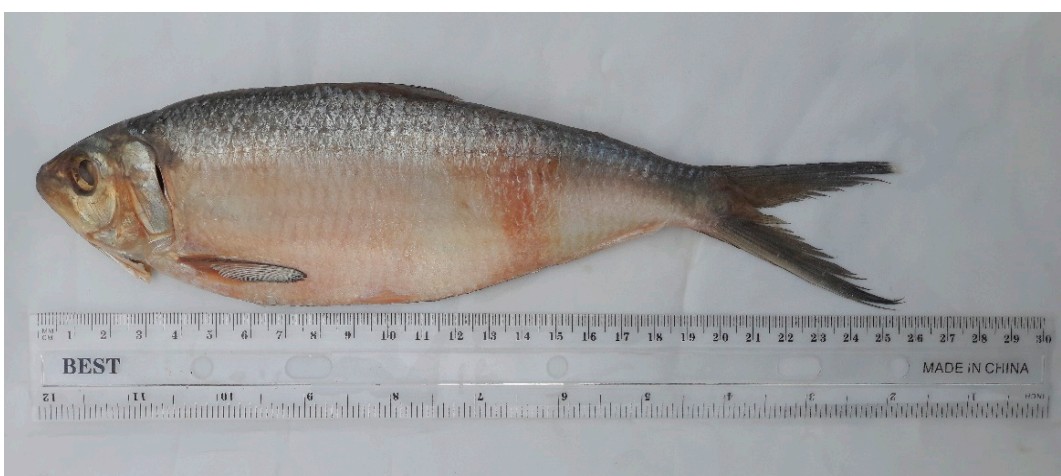

**Figure 4.** *S. maderensis* specimen.

Genetic analysis using DNA barcoding also unambiguously confirmed the identity of the fish species in the study area as *S. maderensis*. The results of the BLAST (Basic Local Alignment Search Tool, from the National Centre for Biotechnology Information (www.ncbi. nlm.nih.gov (accessed on 22 December 2022)) reflect similarities between the search and the NCBI database's biological sequences. The results derived from the analysis using the fish primer [54] indicated that all the samples were *S. maderensis*, with a mean percentage of 98.21% similarity to *S. maderensis* reference sequences in GenBank (Table 4). Hence, the results of the genetic analysis also confirmed that the fish samples were *S. maderensis*.

**Table 4.** Genetic species identification results.

| S/N | Name of Sample | Percentage ID | GenBank Accession No. | BLAST Prediction |
|---|---|---|---|---|
| 1. | OR1 | 99.84 | MT272815.1 | *S. maderensis* |
| 2. | OR2 | 99.69 | MT272814.1 | *S. maderensis* |
| 3. | OR3 | 93.91 | MT272807.1 | *S. maderensis* |
| 4 | OR4 | 98.89 | AP009143.1 | *S. maderensis* |
| 5. | OR5 | 99.21 | AP009143.1 | *S. maderensis* |
| 6. | OR6 | 99.01 | AP009143.1 | *S. maderensis* |
| 7. | LE1 | 89.38 | MT272815.1 | *S. maderensis* |
| 8. | LE2 | 100.0 | MT272815.1 | *S. maderensis* |
| 9. | LE3 | 99.31 | MT272816.1 | *S. maderensis* |
| 10. | LE4 | 99.62 | MT272816.1 | *S. maderensis* |
| 11. | LE5 | 99.83 | MT272811.1 | *S. maderensis* |
| 12. | LE6 | 99.83 | MT272816.1 | *S. maderensis* |

### 4.2. Land Use and Land Cover Change

Land-use change analysis of classified Landsat imageries of the AOI in Ibeju-Lekki revealed extensive changes in the coastal area over the past 36 years. Mangrove forests and cultivated lands declined by approximately 14% and 62%, respectively, between 1984 and 2020, while minor urban development increased by 48%, and land used for major urban/industrial development showed a tremendous increase of about 175% (Table 5).

The analysis in Table 5 indicates dynamic urban growth along the Ibeju-Lekki coastline, which also corroborates the claim by fishers that industrial developments, the Lekki Seaport, and the petrochemical refinery have significantly destroyed coastal habitats and displaced artisanal fishers in Ibeju-Lekki. Also, other urban land uses, such as residential, commercial, recreational, and institutional uses, are major anthropogenic activities that have taken over

the fishing fields and disrupted artisanal fishing in the study area, as seen in the 1984 to 2020 land-cover change maps (Figure 5a–c).

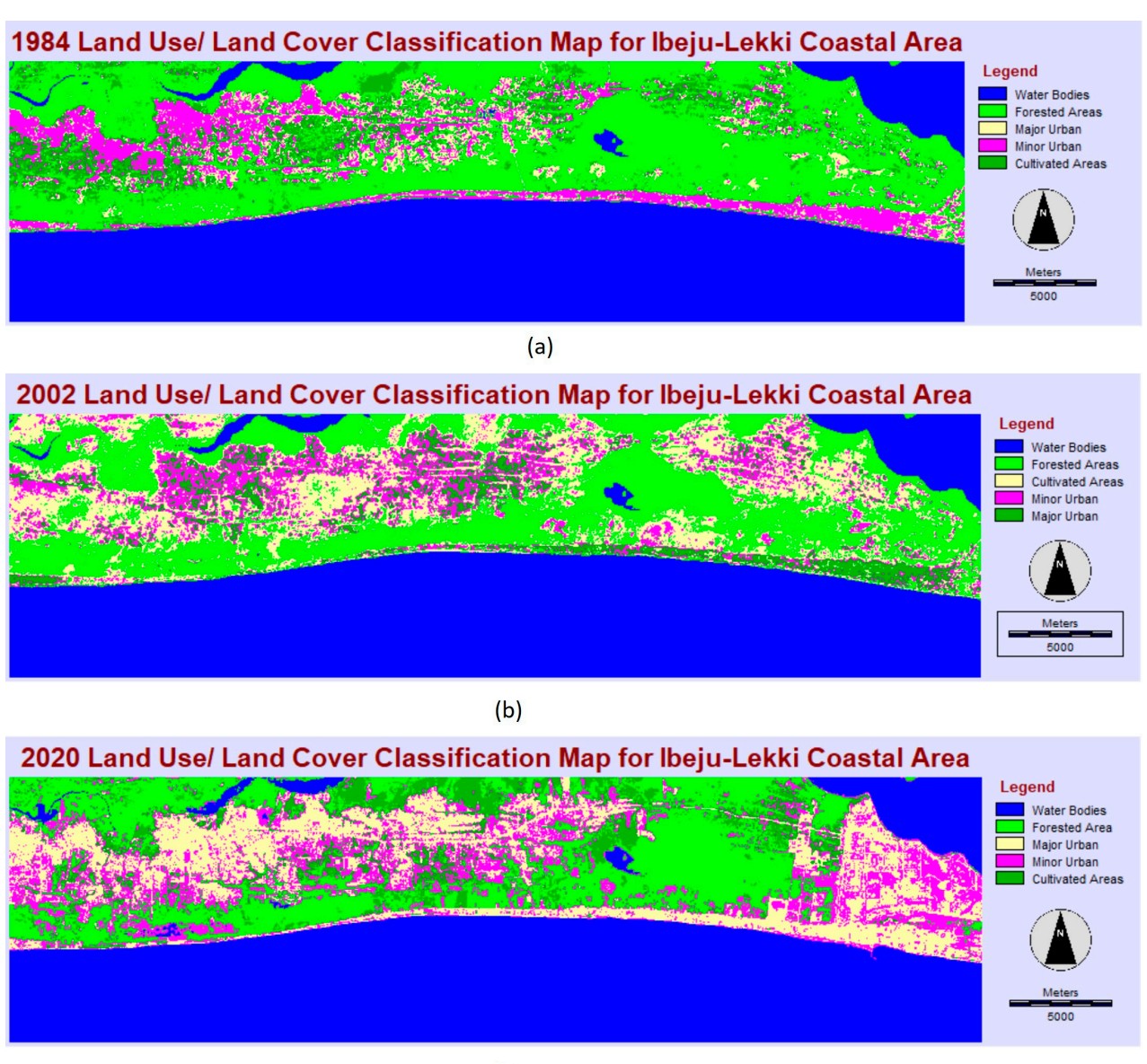

**Figure 5.** The land-use/land-cover change maps of the study area for 1984, 2002, and 2020: (**a**) 1984 land-use/land-cover classification map; (**b**) 2002 land-use/land-cover classification map; (**c**) 2020 land-use/land-cover classification map.

**Table 5.** Land-use change statistics (1984–2020).

| Categories | 1984 (Sq. Km) | % | 2002 (Sq. Km) | % | 2020 (Sq. Km) | % | 1984–2020 Change (Sq. Km) | Change % |
|---|---|---|---|---|---|---|---|---|
| Water Bodies | 268.379 | 45.07 | 268.740 | 45.13 | 267.046 | 44.84 | −1.333 | −0.5 |
| Forested Areas | 136.713 | 22.96 | 187.481 | 31.48 | 118.089 | 19.83 | −18.624 | −13.62 |
| Cultivated Lands | 102.222 | 17.16 | 35.267 | 5.92 | 38.913 | 6.53 | −63.309 | −61.93 |
| Minor Urban | 56.355 | 9.46 | 54.482 | 9.15 | 83.903 | 14.09 | 27.549 | 48.88 |
| Major Urban | 31.868 | 5.35 | 49.568 | 8.32 | 87.585 | 14.71 | 55.717 | 174.84 |
| Total | 595.538 | 100 | 595.538 | 100 | 595.538 | 100 | | |

### 4.3. Water Quality Analyses

The water quality results for the study period revealed that physical and chemical parameters like temperature, pH, salinity, dissolved oxygen, nitrate ($NO_3$), and phosphate ($PO_4$) levels were within tolerable ranges for fish survival, as shown in Table 6. However, the total dissolved solids (TDSs) and biological oxygen demand (BOD) exceeded the recommended levels, which indicated organic pollution from anthropogenic activities [74]. Additionally, the inadequate quantities of chlorophyll-a indicated insufficient food organisms for fish.

**Table 6.** Water quality analysis results for Ibeju-Lekki coastal water.

| Physicochemical Parameters | Min | Max | Mean | Permissible Limits | Remark |
| --- | --- | --- | --- | --- | --- |
| Water Temp. (°C) | 23 | 27 | 25.39 | 24 °C [75] | Within required range |
| pH | 7.32 | 8.06 | 7.88 | 6.5–8.5 [76]; 9.0 [77] | Within required range |
| Salinity (ppt) | 24.86 | 32.17 | 29.37 | Tolerates low salinities [9,12] | Within required range |
| TDS (mg/L) | 11,500 | 30,000 | 24,971.1 | 2000 mg/L [77] | Higher than acceptable level |
| DO (mg/L) | 4.8 | 15.8 | 7.97 | 4.8 mg/L [78] | Within acceptable limit |
| BOD (mg/L) | 0.4 | 11 | 3.37 | 3–6 mg/L—tolerable; 8 mg/L—lethal [79] | Within range/sometimes above the lethal limit |
| $NO_3$ (mg/L) | 0.01 | 2.11 | 0.31 | 20 mg/L [77,80] 10 mg/L [78,81] | Within acceptable limit |
| $PO_4$ (mg/L) | 0.01 | 0.7 | 0.15 | 5 mg/L [77] | Within acceptable limit |
| Chlorophyll-a (μg/L) | 0 | 0.04 | 0.01 | 0.1–8 μg/L [82] | Not up to the required level |

The results of the heavy metal analysis in Table 7 revealed that, though levels found in the study area were above the international standard limits set by the USEPA (1980–2016), the levels of lead, cadmium, iron, manganese, and nickel were within the national limits set by the Federal Environmental Protection Agency (FEPA) in Nigeria [77]. However, chromium levels exceeded the national and international thresholds among the measured heavy metals in Ibeju-Lekki waters. Also, the total petroleum hydrocarbon (TPH) analysis showed that the TPH levels were generally high across all six stations (27.56 mg/L–3985.40 mg/L) when compared to the 10 mg/L permissible limit for coastal waters in Nigeria, indicating hydrocarbon pollution in all the sampled stations throughout the year. Overall, the findings of this study confirmed that coastal waters in Ibeju-Lekki are subject to pollution from industrial effluents and other anthropogenic activities.

**Table 7.** Heavy metal concentrations in water at Ibeju-Lekki.

| Heavy Metals | Min | Max | Mean ± SD | Permissible Limits (USEPA) | Permissible Limits (FEPA, 2003) | Remarks |
| --- | --- | --- | --- | --- | --- | --- |
| Lead (Pb) | 0.00 | 0.93 | 0.20 ± 0.17 | 0.14 [76] | <1.00 | Within acceptable limits nationally but above the international limit |
| Cadmium (Cd) | 0.00 | 0.20 | 0.06 ± 0.06 | 0.03 [83] | <1.00 | |
| Iron (Fe) | 0.31 | 3.16 | 2.62 ± 0.51 | 1.00 [84] | - | |
| Manganese (Mn) | 0.07 | 0.38 | 0.19 ± 0.11 | 0.10 [84] | 5.00 | |
| Nickel (Ni) | 0.21 | 0.93 | 0.64 ± 0.19 | 0.07 [85] | <1.00 | |
| Chromium (Cr) | 0.00 | 7.00 | 1.73 ± 2.09 | 0.18 [86] | <1.00 | Above acceptable limits |

### 4.4. Trends in S. maderensis Abundance

A trend analysis of CPUE (catch per unit effort) showed a consistent decline in the catches of the *S. maderensis* species in Ibeju-Lekki between 2003 and 2019, as shown in Figure 6. The trend line formula is displayed as:

$$y = -0.0072x + 313.52$$

where *y* is the average CPUE and *x* represents time. The slope of the trend line defines the rate of change, which is −0.0072. The negative value of the slope is an indication that there

has been a consistent, gradual decline in CPUE between 2003 and 2019. It also suggests a decline in the efficiency of catching *S. maderensis* over time, with the CPUE decreasing by an average of 0.0072 kg/H.

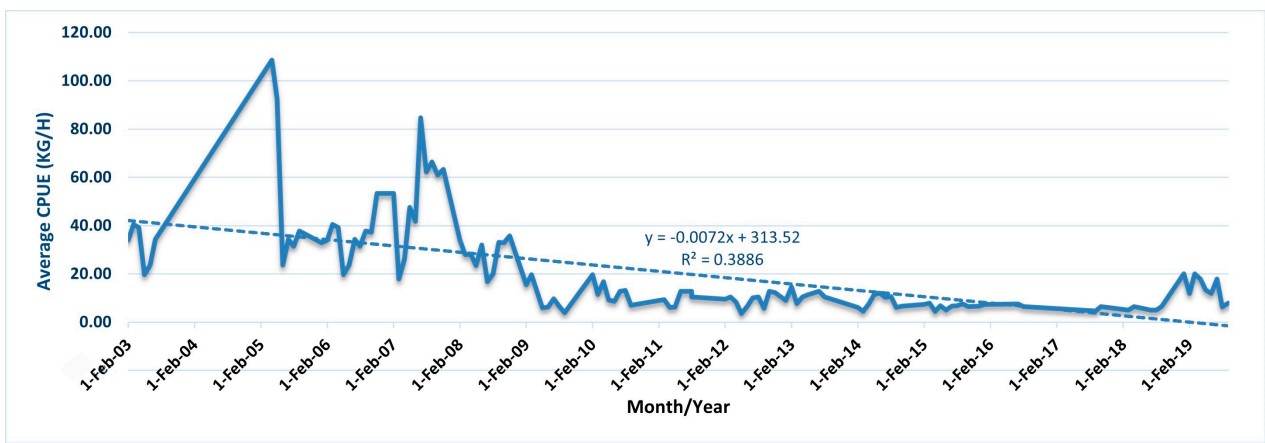

**Figure 6.** Trend analysis of *S. maderensis* abundance.

The decline in *S. maderensis* CPUE shown in Figure 6 suggests behavioural changes in the species, environmental degradation, ecological disturbance, or anthropogenic factors affecting the species and significantly increased fishing efforts [39,87,88]. The graph also shows that the value of $R^2$ is 0.3886, which indicates that 38.86% of the variations in the CPUE are explained by the time trend, which reveals a moderate correlation between time and CPUE. Continuing this trend is unsustainable for the *S. maderensis* fisheries in Ibeju-Lekki.

### 4.5. Anthropogenic Factors Predicting S. maderensis Abundance

Linear regression analysis was conducted to determine anthropogenic factors that predict *S. maderensis* abundance based on fishers' perceptions and experiences. Four independent variables (fishing effort, needed amenities, land use effects, and access to markets) were statistically significant, predicting 40% of the variability in the CPUE (catch per unit effort) of *S. maderensis*, representing abundance.

The results of the linear regression model were significant, as shown in Table 8, as approximately 39.58% ($p < 0.001$) of the variance in CPUE is explainable by V_FISH_EFFORT, V_AMENITY, V_LANDUSE, and V_MKT.

**Table 8.** Linear regression of perceived anthropogenic predictors of *S. maderensis* abundance.

| Variable | *B* | *SE* | 95.00% CI | β | *t* | *p* |
|---|---|---|---|---|---|---|
| (Intercept) | 140.99 | 25.79 | [90.28, 191.71] | 0.00 | 5.47 | <0.001 |
| V_FISH_EFFORT | −7.45 | 0.64 | [−8.72, −6.18] | −0.54 | −11.58 | <0.001 |
| V_AMENITY | −3.23 | 0.55 | [−4.32, −2.15] | −0.27 | −5.85 | <0.001 |
| V_LANDUSE | −0.85 | 0.27 | [−1.39, −0.32] | −0.15 | −3.16 | 0.002 |
| V_MKT | 41.48 | 8.12 | [25.51, 57.45] | 0.22 | 5.11 | <0.001 |

Note. Results: $F_{(4,355)} = 58.14$, $p < 0.001$, $R^2 = 0.40$. Unstandardised regression equation: CPUE $= 140.99 − 7.45 \times$ V_FISH_EFFORT $3.23 \times$ V_AMENITY $− 0.85 \times$ V_LANDUSE $+ 41.48 \times$ V53_MKT.

The regression model indicated that *S. maderensis* fish abundance (CPUE) is significantly predicted by fishing effort (V_FISH_EFFORT), needed amenities (V_AMENITY), land use (V_LANDUSE), and access to markets (V_MKT). Fishing efforts (V_FISH_EFFORT) significantly predicted CPUE negatively, B = −7.45 ($p < 0.001$). This indicates that every unit of fishing effort will decrease CPUE by 7.45 units. Excessive fishing efforts can negatively affect *S. maderensis* fisheries, leading to low yields. The effects of overfishing are corroborated in the literature [89–91]. Needed amenities (V_AMENITY) significantly predicted

CPUE, B = −3.23, (*p* < 0.001), implying that as needs increase by one unit, CPUE decreases by 3.23 units. This means that reducing the needs of fishers by providing good roads, schools, telecommunications, etc., will increase the CPUE. The effect of land use changes (V_LANDUSE) negatively predicts CPUE, B = −0.85, (*p* = 0.002). On average, the effect of one unit of land use change in coastal communities significantly reduces fish abundance by 0.85 units. Conversely, access to markets (V_MKT) significantly enhances abundance (CPUE), B = 41.48 (*p* < 0.001). Access to markets will increase the CPUE by 41.48 units. The linear regression identified four significant predictors of *S. maderensis* abundance: fishing effort, needed amenities, land use effects, and access to markets, which collectively explain 40% of the variability in the CPUE, an indicator of fish abundance.

According to the survey results, most fishers attributed the decline in *S. maderensis* catches to encroachment and loss of fishing grounds due to industrial activities like the petrochemical refinery; critical transportation infrastructures, such as the Lekki Deep Sea Port; and other major urban land uses and anthropogenic activities in Ibeju-Lekki (Figure 7). In addition, 95% of the fishers claimed that their livelihoods and *S. maderensis* fisheries are affected mainly by the refinery's development and other industrial land uses. Meanwhile, 57.5% of the fishers contended that the construction of the Lekki Deep Sea Port has led to a decline in the *S. madenresis* fisheries. Other land use activities like residential, commercial, and recreational developments have a shallow effect on the *Sardinella* spp. fisheries, as these activities affect less than 3% of the fishers, as shown in Figure 8.

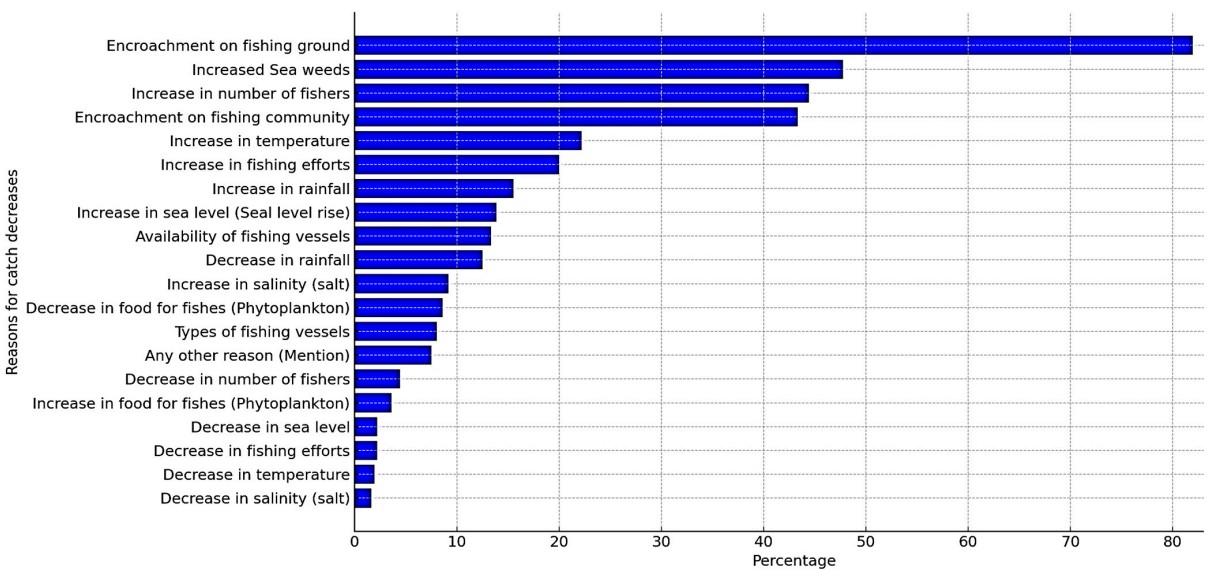

**Figure 7.** Reasons for the decrease in the catches of *S. maderensis*.

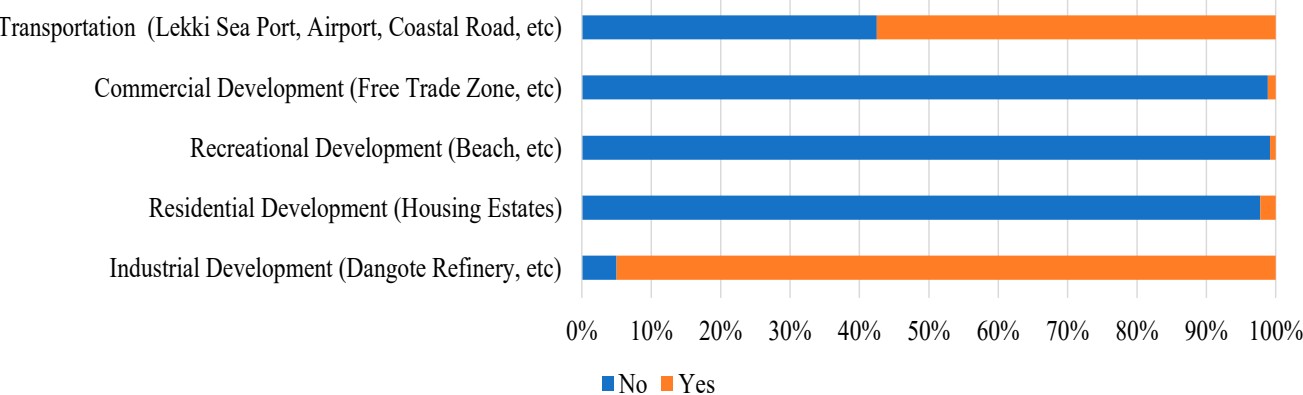

**Figure 8.** Anthropogenic factors (urban land use) affecting *Sardinella* spp. fisheries in Ibeju-Lekki.

In summary, the survey and analyses underscore the significant effects of land use change from once major fishing communities to major urban and industrial development, highlighting the vulnerability of coastal fishing communities. Fishers' perceptions also provide valuable insights into the anthropogenic impacts on the *S. madenresis* fisheries and the need for sustainable coastal management strategies.

## 5. Discussion

This study provides empirical evidence on how growing anthropogenic pressures driven by industrialisation affect the *S. maderensis* fisheries, which coastal communities depend on for their nutrition and livelihoods. The massive land reclamation for constructing the Lekki Deep Sea Port and the petrochemical refinery has extensively damaged the coastal ecosystems in Ibeju-Lekki over time. Effluents from industrial and port activities also significantly pollute the coastal waters, increasing the anthropogenic pressures that threaten small-scale fisheries' sustainability. While these massive developments seem to have economic advantages, they have inadvertently contributed to the decline of the *S. maderensis* fisheries, which are depended upon for nutrition and livelihoods in Ibeju-Lekki. However, several studies have emphasised balancing economic development, environmental conservation, and fisheries sustainability [92–97].

The land-use change analysis of Ibeju-Lekki reveals extensive change to urban/industrial development in the past 36 years, leading to the loss of mangrove and coastal forested areas. The findings of this study align with previous research conducted in other tropical regions, which emphasises the negative impacts of unsustainable coastal development on the resilience of fishing communities [98–100]. Mangroves, seagrass beds, and other coastal habitats are crucial as nursery grounds and provide ecological support for *S. maderensis* fisheries [101,102]. Previous studies show a correlation between declines in fish populations and habitat loss and degradation [103–106]. Consequently, the loss of habitats from anthropogenic activities like dredging, sand filling, construction, and industrial effluents destroys the ecological resources that sustain the *S. maderensis* fisheries in Ibeju-Lekki. Mitigating these adverse impacts requires spatial planning with integrated ecological knowledge and the enforcement of strict environmental management policies to protect fisheries and marine life [107].

Furthermore, the need for stringent environmental policies cannot be overemphasised, as industrial pollution creates a toxic ecosystem for fish survival. The physical and chemical parameter analysis results in Table 6 showed that water temperature, pH, salinity, dissolved oxygen, nitrate ($NO_3$), and phosphate ($PO_4$) levels were still within the ranges tolerable by *S. maderensis* [75,108]. However, the total dissolved solids (TDSs) values ranged between 11,500 mg/L and 30,000 mg/L and were higher than the acceptable limit of 2000 mg/L set by the Federal Environmental Protection Agency. These high TDSs values may affect gill and kidney functions and impact the survival and size of the fish. Mahboob et al. [109] recorded higher values of TDSs in their study of the Arabian Gulf, Saudi Arabia, which was also characterised by prevalent anthropogenic activities similar to those found in Ibeju-Lekki, such as sand dredging, landfilling, and oil spills. The biological oxygen demand (BOD) values occasionally fell within an acceptable range but sometimes exceeded the lethal limit, with values as high as 11 mg/L. Hynes [74] states that BOD values higher than 8 mg/L point to severe pollution. The chlorophyll-*a* levels observed in Table 6 are inadequate, indicating insufficient phytoplankton abundance, potentially adversely affecting the *S. maderensis* populations due to their planktivorous feeding habits. Abdellaoui et al. [110] observed in their studies that changes in chlorophyll-a have a significant impact on sardine abundance, linking minimal chlorophyll-a levels to the gradual decline in yields of sardines in Al Hoceima, South Alboran Sea. Marine phytoplankton are essential not only as food to some fish species but also because they form the foundation of the marine food web [111,112].

The heavy metal analysis results indicate the effects of industrial pollution from the petrochemical refinery and deep sea port activities. These heavy metal analysis results can

be used as baseline values for water quality in Ibeju-Lekki coastal waters because many industries have just started functioning within the last few years. Table 7 reveals that the lead, cadmium, iron, manganese and nickel levels, were within the national limits set by the FEPA in Nigeria but above the international standard limits. Also, the value for chromium in mg/L was above both the local and international standard limits. Previous studies attributed heavy metal pollution to anthropogenic activities in coastal waters [113,114]. For example, lead pollution could be due to oil spills, motorboats, and untreated wastes [114]; other sources include higher concentrations of metals from corrosion in marine construction, landfilling, and construction residuals [109]. The TPH values in the waters of the Ibeju-Lekki communities exceeded the limits of the FEPA standards, which could be attributed to oil spillage in the water [115]. These levels could lead to severe biological and economic impacts on the marine environment, which calls for an urgent regulatory intervention.

The decline in *S. maderensis* CPUE shown in the trend analysis (Figure 6) indicates a declining fish stock and a relatively low fish abundance over time. While a declining trend in CPUE is a common concern in fisheries management, it is often attributed to overfishing or ecosystem degradation [39,116]. However, the $R^2 = 0.3886$ shows that about 61% of the variation in the CPUE is not explained by the time variable alone, suggesting other possible factors like environmental changes, economic or social policy changes, and unsustainable fishing practices. The research also indicates that without a significant shift in fisheries management and conservation practices, the trend may likely continue, aggravating the challenges faced by the marine ecosystem and fishing communities [117].

The fishers' perceptions discussed in Section 4.5 reveal that four anthropogenic factors significantly predict *S. maderensis* abundance. The regression analysis in Table 8 shows that these four independent variables—fishing effort, needed amenities, land use effect, and access to markets—were statistically significant, accounting for 40% of the variability in CPUE, representing fish abundance. This indicates a relatively strong relationship between these anthropogenic factors and rates of catching fish. This finding is consistent with previous findings on the influence of anthropogenic factors on fish stocks, which has been well documented in fisheries science [118–124]. Also, aligning findings from fishers' perceptions with the literature indicated that LEK is crucial in understanding non-biological stressors predicting fish abundance [67,68,71,125].

Fishers in Ibeju-Lekki considered fishing efforts to have a negative relationship with fish abundance. Table 8 shows that CPUE decreases by 7.45 units for every unit increase in the fishing effort variable (V_FISH_EFFORT). This result corroborates the literature on the effect of overfishing on fish abundance [39,117,126]. This result requires that fishing efforts be managed to avoid overexploitation and ensure the sustainability of the *S. maderensis* fisheries. Previous studies suggested that controlled fishing practices and fisheries management systems, grounded on rights-based principles that co-opt fishers in the management process, could help mitigate overfishing and empower fishing communities effectively [39,127].

Furthermore, this research corroborates previous studies asserting that land use and habitat alterations can significantly impact marine ecosystems and fish abundance [118,119,128]. The results in Table 8 show that the land-use effect variable (V_LANDUSE) is a significant predictor of fish abundance, with a negative coefficient of $-0.85$ ($p = 0.002$), indicating that the ongoing massive coastal developments like the petrochemical refinery, the deep sea port, and other significant urban developments have a negative impact on the *S. maderensis* fisheries. The results underscore the need for sustainable marine spatial planning (MSP).

In addition, the government and stakeholders should augment MSP by providing needed amenities and market access. On the one hand, needed amenities (V_AMENITY) negatively predicts CPUE with a regression coefficient (B) value of $-3.23$ ($p < 0.001$), indicating that for every unit of deficiency of needed amenities, the CPUE decreases by 3.23 units, implying that providing needed amenities in fishing communities will enhance the fishing efficiency and fish abundance. On the other hand, access to markets (V_MKT) is the only positive predictor of CPUE, with a B value of 41.48 ($p < 0.001$), indicating that fish

abundance is strongly influenced by better access to markets. The results also reinforce the importance of amenities and market access as drivers of small-scale fisheries. This study also resonates with studies by Bene et al. [129,130] on the role of socioeconomic drivers in sustaining small-scale fisheries.

The findings generally emphasise the necessity of adopting a comprehensive and inclusive strategy for coastal development that safeguards the fundamental natural resources that sustain the local population's livelihoods [131]. Presently, existing development plans and policies for Ibeju-Lekki emphasise top-down development strategies prioritising industrial enterprises, aiming to achieve economic benefits. However, these policies overlook the socio-ecological impacts on marginalised small-scale fisheries [132]. Nevertheless, neglecting these consequences undermines long-term sustainability. Hence, an ecosystems-based approach in fisheries management and an integrated marine spatial planning strategy for land and water use are needed to conserve vital fish habitats [133,134]. Implementing rigorous water quality monitoring and strict enforcement of effluent standards is crucial for mitigating industrial pollution, and controlling detrimental activities such as unregulated sand dredging, which harms resilience, is also necessary. Additionally, the implementation of fisheries management systems based on rights-based principles has the potential to mitigate overfishing and empower fishing communities effectively [135]. In general, mitigating anthropogenic threats necessitates the implementation of multi-level governance approaches that effectively balance economic development, ecological sustainability, and social equity [136].

This investigation provides a framework for research approaches to assess anthropogenic threats in small-scale fisheries in developing countries. In summary, this research highlights the importance of addressing human-induced challenges such as pollution and habitat loss to ensure the sustainability of small-scale fisheries, which play a vital role in providing employment, food security, and nutrition in rapidly developing coastal regions.

## 6. Conclusions

This research investigated the effects of anthropogenic activities on small-scale fisheries, which play a crucial role in supporting the livelihoods of economically disadvantaged populations in the coastal regions of Nigeria. The study highlights threats posed by pollution, extensive habitat loss, and degradation due to rapid urbanisation and industrialisation over the past few decades in Ibeju-Lekki, leading to the depletion of mangrove ecosystems, which are crucial breeding grounds for fish like *S. maderensis*. The findings underscore the urgent need for inclusive fisheries management and sustainable coastal development strategies to safeguard these vital ecosystems and dependent fishing communities to foster the resilience and conservation of the *S. maderensis* fish species. High levels of hydrocarbon pollution and heavy metals from industrial effluents create a toxic marine environment detrimental to the productivity and survival of small pelagic species like *S. maderensis*. The significant decline in fish abundance over the past two decades, coupled with the fishers' perceptions, indicates the adverse impact of habitat loss through land use change, overfishing, lack of needed amenities, and the need for economic opportunities for fishing communities that support local livelihoods and food security.

This study contributes to the growing body of evidence calling for a sustainable and inclusive approach to coastal and marine resource management, especially in the Global South and the need for MSP, stricter pollution control, habitat restoration, and biodiversity conservation. The study combines qualitative and quantitative approaches with ecosystems-based management strategies to understand the multifaceted impacts of anthropogenic activities on small-scale fisheries. Also, the study addresses the existing data gap respecting *S. maderensis* fisheries in the coastal regions of Nigeria, which have been previously identified by the FAO [44], thereby giving critical insights into the pros and cons of sustainable fisheries management in similar contexts. Applying genetic and morphological techniques to identify the *Sardinella* spp. species exploited in the region enhanced the accuracy of our research findings. It also provided a better scientific under-

standing of the current status of the *Sardinella* spp. stock in Ibeju-Lekki coastal waters. In addition, the research methods and findings provide a foundation for future studies in the fisheries sustainability domain, where incorporating scientific evidence and the LEK of fishing communities into ecosystems-based management strategies can contribute to small-scale fisheries in the coastal region, thereby attaining SDG 14.

Future investigations should broaden the spatial and temporal scope by employing ecosystem modes to unearth the complexities regarding small-scale fisheries dynamics and anthropogenic impacts for more insights into strategies for mitigation and adaptation. We suggest an inclusive multi-stakeholder approach to fisheries management involving the state, fishing communities, non-governmental organisations, and the private sector. Policies should prioritise the protection of habitat and marine resources, pollution control, regulation of fishing efforts, amenities provision, market development, and economic incentives for fishing communities. This research has shed light on anthropogenic impacts affecting small-scale *S. maderensis* fisheries in the coastal area of Nigeria. Implementing the recommendations will require concerted efforts that align with SDG 14. Incorporating empirical evidence and local ecological knowledge into the ecosystems-based management of fisheries will birth resilient small-scale *S. maderensis* fisheries supporting sustainable livelihoods, nutritional well-being, and biodiversity conservation in rapidly developing coastal areas.

**Author Contributions:** Conceptualisation, T.A.; methodology, T.A.; software, T.A. and S.D.; validation, D.A., I.O. and O.S.; formal analysis, T.A. and S.D.; investigation, T.A.; resources, D.A., I.O. and O.S.; writing—original draft preparation, T.A.; writing—review and editing, T.A. and S.D.; supervision, D.A., I.O. and O.S. All authors have read and agreed to the published version of the manuscript.

**Funding:** This research and APC were funded by the Africa Centre of Excellence in Coastal Resilience (ACECoR), University of Cape Coast, with support from the World Bank and the Government of Ghana. The World Bank ACE grant number is credit number 6389-G.

**Institutional Review Board Statement:** This study was reviewed a reviewed as part of the first author's PhD thesis nd approved by the Institutional Review Board of the University of Cape Coast, with the reference number UCCIRB/CANS/2020/09, and the Health Research Ethics Committee of the College of Medicine of the University of Lagos (HRECMUL), Lagos, Nigeria, with the reference number CMUL/HREC/10/20/784.

**Informed Consent Statement:** Informed consent was obtained from all subjects involved in the study.

**Data Availability Statement:** The data presented in this research are available upon request from the corresponding author.

**Acknowledgments:** The authors acknowledge the World Bank Africa Centre of Excellence in Coastal Resilience (ACECoR); the University of Cape Coast, Ghana; the Association of African Universities; and the Government of Ghana for funding this research and the Ibeju-Lekki fishers in the study area who voluntarily took part in the survey interviews.

**Conflicts of Interest:** The authors declare no conflicts of interest.

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
