# Peer review of "Effects of Anthropogenic Activities on Sardinella maderensis (Lowe, 1838) Fisheries in Coastal Communities of Ibeju-Lekki, Lagos, Nigeria"

_sustainability, doi:10.3390/su16072848_

Round 1

Reviewer 1 Report

Comments and Suggestions for Authors

The authors present a detailed analysis of how anthropogenic activities have affected Sardinella maderensis fisheries in in coastal communities of Ibeju-Lekki. The results look a little mixed up. We know that fishing is an important factor causing the decline of fisheries, but the author only mentions it briefly, and this part needs further analysis. In addition, there are some minor issues in the paper that need to be modified by the author.

1. Lines 24 and 285: p<0.001, R2=0.05?

2. Lines 56, 61, 71: Sardinella maderensis should be S. maderensis. Please check other text and make sure Latin names need to be abbreviated after their first appearance.

3. Lines 82: S. maderensis should be italic. Please check other text and make sure Latin names need to be italic.

4. Figure 5 is not clear enough and “Reasons attributed to catch decrease” is redundant.

5. The conclusion did not summarize the results of the paper well, and the results in the paper do not mention climate change

Author Response

We sincerely appreciate your insightful comments on our manuscript. We have carefully reviewed the comments and believe they will significantly improve our manuscript. Find below a point-by-point response to all concerns raised:

Comment Response
The authors present a detailed analysis of how anthropogenic activities have affected Sardinella maderensis fisheries in in coastal communities of Ibeju-Lekki. The results look a little mixed up. We know that fishing is an important factor causing the decline of fisheries, but the author only mentions it briefly, and this part needs further analysis. In addition, there are some minor issues in the paper that need to be modified by the author.

Results have been improved. We have also improved the content on fishing efforts with further analysis, as suggested.

The issues mentioned in the attached paper have been addressed.

1. Lines 24 and 285: p<0.001, R2=0.05? Updated already
2. Lines 56, 61, 71: Sardinella maderensis should be S. maderensis. Please check other text and make sure Latin names need to be abbreviated after their first appearance. All corrected
3. Lines 82: S. maderensis should be italic. Please check other text and make sure Latin names need to be italic. All corrected
4. Figure 5 is not clear enough and “Reasons attributed to catch decrease” is redundant. Updated with clear figure and text. (See Figure 7 in the revised manuscript) 
5. The conclusion did not summarize the results of the paper well, and the results in the paper do not mention climate change The conclusion has been improved as noted.

We believe that revisions will significantly improve our manuscript's quality, and we appreciate the reviewer's guidance. We look forward to submitting our revised manuscript.

Reviewer 2 Report

Comments and Suggestions for Authors

1- some figures and tables not mentioned in paragraph.

2- there are mistakes in mention figures and tables.

3- name of species must be write with Italic font.

In attachment file , I highlighted on mistakes  

Author Response

We sincerely appreciate your insightful comments on our manuscript. We have carefully reviewed the comments and believe they will significantly improve our manuscript. Find below a point-by-point response to all concerns raised:

Comments Responses
1- some figures and tables not mentioned in paragraph. All issues have been addressed in the revised manuscript
2- there are mistakes in mention figures and tables. All tables and figures corrected
3- name of species must be write with Italic font All written in italic font
In attachment file , I highlighted on mistakes   Issues raised in the attached file has been addressed.

The suggested revisions will immensely improve the quality of our manuscript, and we appreciate the reviewer's guidance. We look forward to submitting the revised manuscript for your consideration.

Sincerely,

Temi Adewale

Reviewer 3 Report

Comments and Suggestions for Authors

This paper examines the impact of anthropogenic activities on Sardinella madere through a combination of qualitative and quantitative research methods. The study holds significance for the sustainable development of local fisheries, but the article itself poses challenges for readers due to unclear explanations in many sections. The paper have to been modified.

(1) The research employs two methods, but the logical flow of these methods lacks clarity. It would be beneficial to enhance comprehension by incorporating a flowchart.

(2) The author asserts that anthropogenic activities influence S. maderensis fisheries, studying these effects through a linear regression model. The primary concern lies in quantifying the impact of these activities. The paper lacks pertinent information on land use change and amenities. Proper elucidation of the calculation for land use change and amenities, along with the presentation of their distribution, is essential.

(3) The introduction of linear regression is insufficient, and the construction of each variable's data remains unexplained. For instance, the effort data is a continuous variable, while land data from a remote sensing map is discontinuous, with only three times sample points. A dedicated section should be included to address these issues.

(3)Typically, there exists a positive correlation between fishing yield and fishing effort, initially positive and later weakly negative, or even displaying a concave distribution. The effects of variables should be elaborated upon in the discussion section.

(4)The paper lacks the display of fish sample and water sample locations. In Ibeju-Lekki, two landing locations, Orimedu and Badore, are mentioned, or possibly just Orimedu. However, the meaning of 'LE' is unclear. It is imperative to provide full names when introducing abbreviations.

(5)The purpose of water quality analysis in the paper is unclear. Although seemingly excluded from the linear regression model, the discussion predominantly revolves around it. To enhance clarity, avoid discussing within the results section; instead, address results in the discussion section.

The subscript in Figure 5 is not clearly legible.

Author Response

We sincerely appreciate your insightful comments on our manuscript. We have carefully reviewed the comments and believe they will significantly improve our manuscript. Find below a point-by-point response to all concerns raised:

Comments Response
This paper examines the impact of anthropogenic activities on Sardinella madere through a combination of qualitative and quantitative research methods. The study holds significance for the sustainable development of local fisheries, but the article itself poses challenges for readers due to unclear explanations in many sections. The paper have to been modified. The article has been modified as suggested.
(1) The research employs two methods, but the logical flow of these methods lacks clarity. It would be beneficial to enhance comprehension by incorporating a flowchart. We are considering this suggestion.
(2) The author asserts that anthropogenic activities influence S. maderensis fisheries, studying these effects through a linear regression model. The primary concern lies in quantifying the impact of these activities. The paper lacks pertinent information on land use change and amenities. Proper elucidation of the calculation for land use change and amenities, along with the presentation of their distribution, is essential. Table 3 in the reworked article defines the variables used for the regression analysis. The updated article elucidates pertinent information and calculations made from the observed variables.
(3) The introduction of linear regression is insufficient, and the construction of each variable's data remains unexplained. For instance, the effort data is a continuous variable, while land data from a remote sensing map is discontinuous, with only three times sample points. A dedicated section should be included to address these issues. The land use effect in the linear regression was derived from fishers' perception data (See table 3). Data on fishing efforts also was derived from the observed variable. There were two fishing efforts data: a) landing data collected by LASDA (2003-2019) and b) fishing efforts calculated from observed variables as stated in Table 3
(3)Typically, there exists a positive correlation between fishing yield and fishing effort, initially positive and later weakly negative, or even displaying a concave distribution. The effects of variables should be elaborated upon in the discussion section. We have discussed the fishing effort based on our findings.
(4)The paper lacks the display of fish sample and water sample locations. In Ibeju-Lekki, two landing locations, Orimedu and Badore, are mentioned, or possibly just Orimedu. However, the meaning of 'LE' is unclear. It is imperative to provide full names when introducing abbreviations We will add supplementary data on the GPS-recorded locations of the fish/water samples. However, Figure 2 shows the six foremost landing stations used as fish/water sample locations.
(5)The purpose of water quality analysis in the paper is unclear. Although seemingly excluded from the linear regression model, the discussion predominantly revolves around it. To enhance clarity, avoid discussing within the results section; instead, address results in the discussion section. The purpose of water quality analysis has been clarified in the paper. The linear regression used fishers' perception data only. As suggested, we have ensured discussions in the result section are avoided.
The subscript in Figure 5 is not clearly legible. A clear subscript has been added to Figure 5 in the updated version of the manuscript.

We believe the revisions will significantly improve the quality of our manuscript, and we appreciate the reviewer's guidance. We look forward to submitting the revised paper for your consideration.

Sincerely,

Temi Adewale

Reviewer 4 Report

Comments and Suggestions for Authors

My comments:
1. The topic of this paper is interesting and innovative and it will contribute in related research field.

2. A section of “Related Works” or “Literature Review” is necessary for this paper.

3. The section of “2. Materials and Methods” and “3. Results” are well-written.

4. “3.1. Description of the Study Area” should be corrected as “2.1. Description of the Study Area”.

5. The section of Conclusions” must be reinforced more. For example, the contributions to academic research as well as theoretical implications and research limitations.

Comments on the Quality of English Language

Minor editing of English language required

Author Response

We sincerely appreciate your insightful comments on our manuscript. We have carefully reviewed the comments and believe they will significantly improve our manuscript. Find below a point-by-point response to all concerns raised:

We believe the revisions will significantly improve the quality of our manuscript, and we appreciate the reviewer's guidance. We look forward to submitting the revised paper for your consideration.

Temi Adewale

Comments Responses
1. The topic of this paper is interesting and innovative and it will contribute in related research field. Note
2. A section of “Related Works” or “Literature Review” is necessary for this paper. Section 2. "Literature review" has been added
3. The section of “2. Materials and Methods” and “3. Results” are well-written. This has been done.
4. “3.1. Description of the Study Area” should be corrected as “2.1. Description of the Study Area”. This has been corrected.
5. The section of Conclusions” must be reinforced more. For example, the contributions to academic research as well as theoretical implications and research limitations. The conclusion has been reinforced.

Reviewer 5 Report

Comments and Suggestions for Authors

The manuscript contributes various, valuable but unsorted information, concerning a target fish species:

data about morphology and genetics of a given stock, combined with time series of land use and land cover change; its fisheries; fishermen inquiries; only recent water analyses.

In general, the manuscript pretends to be an action plan, rather than a scientific paper. There are too many facts joined together, that is almost impossible to derive clear connections.

-        What is the purpose of morphologic and genetic determination? How this is connected to the population’s decline? Genetic homogeneity of a particular stock can be connected to a conservative genome, as well as a bottleneck (population collapse due to overfishing).

These data should be published separately, in order to describe this particular fish stock, an important contribution to the species’ assessment.

-        How time series data of fisheries and land use are connected with recent water analyses? Population declines could be connected with single-pressure events (e.g., acute oil splits) or chronic pressure (overfishing, constant pollution). In this case, such a correlation as the declared cannot be derived.

- Fishermen’s questionnaires reflect their theses but contribute unproven facts. How reliability was measured? Their beliefs have to be evaluated by certain measurements, e.g., changes in fishing effort, salinity, rainfall, etc.

This material is better to be included in a separate paper.

-        The discussion is parted mainly by general statements, not directly connected to the study.

Minor comment: the figure and table legends have to be more precise and informative.

Under these circumstances, the authors are encouraged to split the study into smaller parts, to improve the analyses and the discussion, and to re-submit the material.

For additional comments, please see the attached file.

Author Response

We sincerely appreciate your insightful comments on our manuscript. We have carefully reviewed the comments and believe they will significantly improve our manuscript. Find below a point-by-point response to all concerns raised:

Comments Responses
The manuscript contributes various, valuable but unsorted information, concerning a target fish species: Noted. We have tried to sort out as much as possible
Data about morphology and genetics of a given stock, combined with time series of land use and land cover change; its fisheries; fishermen inquiries; only recent water analyses. Noted.
In general, the manuscript pretends to be an action plan, rather than a scientific paper. There are too many facts joined together, that is almost impossible to derive clear connections. We have tried to show the linkages between the various aspects covered in the paper. We have also removed some aspects and reinforced some that were not previously explicit.

What is the purpose of morphologic and genetic determination? How this is connected to the population’s decline? Genetic homogeneity of a particular stock can be connected to a conservative genome, as well as a bottleneck (population collapse due to overfishing).

These data should be published separately, in order to describe this particular fish stock, an important contribution to the species’ assessment.

Species identification was necessary because the target species have been rarely studied in the region. This will help reinforce the findings about the decline in this stock exploited in the region. It also validates the IUCN and FAO's claim of being a vulnerable species. Moreover, we want to establish if anthropogenic factors predict the abundance of the target species.
How time series data of fisheries and land use are connected with recent water analyses? Population declines could be connected with single-pressure events (e.g., acute oil splits) or chronic pressure (overfishing, constant pollution). In this case, such a correlation as the declared cannot be derived.

The time series data establishes the stock's decline based on data collected from government agencies. Land use/land cover change analysis confirms that rapid urbanisation has led to habitat loss with the loss of mangrove forests in the coastal areas. The coastal communities are losing fishing grounds to urban/industrial/transport land uses.

However, water analysis is used to ascertain the presence of toxic pollutants to the fish in the Ibeju-Lekki fishing waters.

Fishermen’s questionnaires reflect their theses but contribute unproven facts. How reliability was measured? Their beliefs have to be evaluated by certain measurements, e.g., changes in fishing effort, salinity, rainfall, etc.

This material is better to be included in a separate paper.

The fishers' questionnaire was tested for reliability through a pilot survey, and the Cronbach Alpha value was 0.823, which shows reliability. In addition, local ecological knowledge derived from fishers' perceptions helps support scientific findings and has been used by several authors, as cited in the updated manuscript.

 The discussion is parted mainly by general statements not directly connected to the study.

Noted. We have been more specific to the study's findings in the updated version. 

Minor comment: the figure and table legends must be more precise and informative.

We have corrected the figures and tables to reflect precise information
Under these circumstances, the authors are encouraged to split the study into smaller parts to improve the analyses and the discussion and to re-submit the material. We have tried to analyse and discuss in smaller sections
For additional comments, please see the attached file. We have tried to address all issues raised in the attached document

We believe the revisions will significantly improve the quality of our manuscript, and we appreciate the reviewer's guidance. We look forward to submitting the revised paper for your consideration.

Temi Adewale

Round 2

Reviewer 1 Report

Comments and Suggestions for Authors

The author responded seriously to my concerns. I would suggest that Figure 6 could be made more aesthetically pleasing.

Author Response

Thank you for your immense contribution to improving our manuscript. We agree that Figure 6 should be enhanced. We have enhanced it to be visually pleasing in the updated manuscript. Thank you.

Reviewer 3 Report

Comments and Suggestions for Authors

The revised manuscript has been improved and all the suggestion were accpeted .

Author Response

Thank you for your immense contributions to improving the manuscript.

Reviewer 5 Report

Comments and Suggestions for Authors

The MS has been essentially improved, according to most comments. Nevertheless, there is still temporal mismatch between the various data, incorporated in the analyses. Catch data begin from 2003, land use data from 1984, and water analysis from 2021. Please unify!

Author Response

Thank you for your immense contribution to improving our manuscript. On the mismatch observed, we agree it would have been the best thing to have the data starting from the exact dates; however, due to poor fishery data collection in the study area, we are contained to use the data available (from 2003) obtained from the repositories of government agencies in Lagos. The water analysis data were collected from 2021 because that was when the research started (a PhD project); there has not been longitudinal data on water analysis for the study area since these have not really been investigated. However, we have suggested comprehensive and regular water analysis for the coastal waters for sustainability.